# Exploring the Role of Peripheral Macrophages in Glioma Progression: The Metabolic Significance of Cyclooxygenase-2 (COX-2)

**DOI:** 10.3390/ijms26136198

**Published:** 2025-06-27

**Authors:** Jens Pietzsch, Magali Toussaint, Cornelius Kurt Donat, Alina Doctor, Sebastian Meister, Johanna Wodtke, Markus Laube, Frank Hofheinz, Jan Rix, Winnie Deuther-Conrad, Cathleen Haase-Kohn

**Affiliations:** 1Department of Radiopharmaceutical and Chemical Biology, Institute of Radiopharmaceutical Cancer Research, Helmholtz-Zentrum Dresden-Rossendorf (HZDR), 01328 Dresden, Germany; j.pietzsch@hzdr.de (J.P.); a.doctor@hzdr.de (A.D.); s.meister@hzdr.de (S.M.); j.wodtke@hzdr.de (J.W.);; 2School of Science, Faculty of Chemistry and Food Chemistry, Technische Universität Dresden, 01062 Dresden, Germany; 3Department of Neuroradiopharmaceuticals, Institute of Radiopharmaceutical Cancer Research, Helmholtz-Zentrum Dresden-Rossendorf, 04318 Leipzig, Germany; m.toussaint@hzdr.de (M.T.); w.deuther-conrad@hzdr.de (W.D.-C.); 4Department of Medicinal Radiochemistry, Institute of Radiopharmaceutical Cancer Research, Helmholtz-Zentrum Dresden-Rossendorf, 01328 Dresden, Germany; c.donat@hzdr.de; 5Department of Positron Emission Tomography, Institute of Radiopharmaceutical Cancer Research, Helmholtz-Zentrum Dresden-Rossendorf, 01328 Dresden, Germany; f.hofheinz@hzdr.de; 6Department of Medical Physics and Biomedical Engineering, Faculty of Medicine Carl Gustav Carus, TU Dresden, Fetscherstrasse 74, D-01307 Dresden, Germany; jan.rix@tu-dresden.de

**Keywords:** CRISPR/Cas9, spheroid model, tumor xenografts, sessile macrophages, bone marrow, tumor microenvironment, prostaglandins

## Abstract

Glioblastoma (GBM) is the most aggressive form of malignant gliomas, with the eicosanoid-synthesizing enzyme cyclooxygenase-2 (COX-2) playing a pivotal role in its progression via the COX-2/prostaglandin E2/4 axis. COX-2 upregulations in tumor cells induces a pro-inflammatory tumor microenvironment (TME), affecting the behavior of invading bone marrow-derived macrophages (Mϕ) and brain-resident microglia (MG) through unclear autocrine and paracrine mechanisms. Using CRISPR/Cas9 technology, we generated COX-2 knockout U87 glioblastoma cells. In spheroids and in vivo xenografts, this resulted in a significant inhibition of tumorigenic properties, while not observed in standard adherent monolayer culture. Here, the knockout induced a G1 cell cycle arrest in adherent cells, accompanied by increased ROS, mitochondrial activity, and cytochrome c-mediated apoptosis. In spheroids and xenograft models, COX-2 knockout led to notable growth delays and increased cell death, characterized by features of both apoptosis and autophagy. Interestingly, these effects were partially reversed in subcutaneous xenografts after co-culture with Mϕ, while co-culture with MG enhanced the growth-suppressive effects. In an orthotopic model, COX-2 knockout tumors displayed reduced proliferation (fewer Ki-67 positive cells), increased numbers of GFAP-positive astrocytes, and signs of membrane blebbing. These findings highlight the potential of COX-2 knockout and suppression as a therapeutic strategy in GBM, particularly when combined with suppression of infiltrating macrophages and stabilization of resident microglia populations to enhance anti-tumor effects.

## 1. Introduction

Glioblastoma (GBM) is the most aggressive primary brain cancer, with a median survival of only 8 to 18 months, despite typically agressive surgical resection, chemo- and radiotherapy [1,2]. A crucial factor contributing to GBM progression and invasiveness is the tumor microenvironment (TME), a dynamic ecosystem of both neoplastic and non-neoplastic cells [3,4]. Among these, tumor-associated macrophages (TAMs) are par-ticularly prominent, comprising 40–50% of the tumor mass and correlating with higher tumor grade and shorter patient survival [5]. TAMs encompass brain-resident microglia (MG) and bone marrow-derived macrophages Mϕ, both playing pivotal roles in GBM pathogenesis [4].

A key driver within the TME are different inflammatory processes, heavily mediated by TAMs [6]. At the molecular level, the Cyclooxygenase-2 (COX-2)-prostaglandin E2/E4 (PGE2/PGE4) axis emerges as a critical pathway [6,7,8]. COX-2 expression is induced in TAMs and constitutively expressed in human adherent GBM cell lines, correlating with tumor grade and poor prognosis [9,10,11]. Interestingly, this pathway is predominantly lo-calized in hypoxic areas, where hypoxia-inducible factor-1α (HIF-1α) further enhances COX-2 expression [11,12,13]. Hypoxia not only seems to drive COX-2 upregulation, but also modulates TAM-tumor cell communication through altered ligand-receptor interactions, thereby promoting tumor progression. Additionally, hypoxia and associated oxidative stress amplify inflammation, reactive oxygen species (ROS) production, and apoptosis, further increasing tumor aggressiveness [14,15,16].

Given the critical role of the COX-2/PGE2 axis in GBM, therapeutic targeting of this pathway offers potential benefits, including reduced cell proliferation, angiogenesis, and enhanced apoptosis [17,18]. However, translating these findings into clinical strategies requires appropriate models to dissect the complexity of TAM-GBM interactions.

In this study, we aim to further elucidate the role of COX-2 in the TAM-GBM dynamic TEM. Employing genetically modified human glioblastoma U87 cells with a complete COX-2 knockout (U87^COX-2KO^), we investigated metabolic interactions with TAM popula-tions, including bone marrow-derived macrophages and microglia. Comparisons were made with U87 wild-type cells (U87^wt^) in adherent cultures, spheroids, and corresponding heterotopic xenograft mouse models. Finally, to better evaluate the influence of the COX-2/PGE2 axis on direct TAM-tumor interactions and tumor growth, an appropriate orthotopic xenotransplantation model was employed.

## 2. Results

### 2.1. In Vitro Experiments in Adherent Cell Culture

The cell growth rate of U87^wt^ and U87^COX−2KO^ cells under normoxia and a sufficient supply of nutrients was not statistically different (Figure 1a). The stable COX-2 knockout in U87 cells, confirmed with RFP flow cytometry (Appendix A), resulted in a significant increase in G1 cell cycle arrest under normoxic (* *p* < 0.05) and hypoxic conditions compared to U87^wt^ cells (Figure 1b; Appendix A).

Antibody array analysis reveals an upregulation of apoptosis-related proteins (e.g., MAPK15 and p53) and downregulation of tumor–TME crosstalk markers (e.g., Il-10, CXCL2, and E-Selectin) in COX-2 knockout cells (Figure 1c). PGE2 levels were unaffected by COX-2 knockout, suggesting a COX-1-mediated compensatory pathway during arachidonic acid metabolism (Appendix A). Clonogenic survival analysis after X-ray radiation indicated significantly reduced plating efficiency in U87^COX−2KO^ at higher radiation doses (4–10 Gy) (Appendix A).

In monolayer cultures, COX-2 knockout significantly increases the number of meshes and segments (* *p* < 0.05), as demonstrated by the angiogenesis network formation on the hCMEC endothelial cells (Appendix A). Interestingly, U87^COX−2KO^ shows significantly increased cell migration (up to 72 h, ** *p* < 0.01) and invasion (up to 84 h, *** *p* < 0.001) (Appendix A). This trend persisted under nutrient deprivation and after stimulation with activated macrophage supernatants, suggesting an elevated response to inflammatory stimuli (Appendix A).

Western blot analysis confirms the successful COX-2 knockout in U87^COX−2KO^ cells (Figure 1d). No difference in COX-1 expression was observed between U87^wt^ and U87^COX−2KO^ cells, indicating that COX-2 knockout did not trigger the compensatory upregulation of COX-1. Similarly, monocytes (MCs) and microglia (MG), lacking COX-2 expression, showed no change in COX-1 levels, and reduced expression for MG. Interestingly, lysyl oxidase (LOX), a protein associated with tumor progression, was reduced in U87^COX-2KO^ and MG cells. Additionally, U87^COX-2KO^ and MC cells exhibited significantly higher cytochrome c release from mitochondria, compared to wildtype cells.

COX-2 knockout cells showed significantly reduced basal mitochondrial activity (Figure 1e, upper panel), but this increased under stress conditions, such as nutrient deprivation or inflammatory stimulation.

ROS activity (Figure 1e, middle panel) and malondialdehyde (MDA) (Figure 1e, lower panel) levels were consistently found to be significantly higher in COX-2 knockout cells across all tested conditions, including oxidative stress (H_2_O_2_) and hypoxia-inducing treatments (CoCl_2_).

Under normoxic conditions, U87^COX−2KO^ cells exhibited significantly reduced glucose (*** *p* < 0.001) (Figure 1f, upper left panel) and significantly increased [^18^F]FMISO uptake (*** *p* < 0.001) (Figure 1f, upper right panel). Together, this indicates higher intrinsic hypoxia in comparison to U87 wildtype cells.

When U87^wt^ cells were co-cultured with microglia, a significantly reduced [^18^F]FMISO radiotracer uptake was observed. All other conditions inducing either oxidative, hypoxic, or nutritional deprivation were not affected by being co-cultured with microglia (Figure 1f, lower panel left). U87^COX−2KO^ cells also responded with reduced [^18^F]FMISO uptake to being co-cultured with microglia under all stress conditions (****p* < 0.001) (Figure 1f, lower right panel). COX-2 knockout cells displayed higher hypoxia under stress compared to U87^wt^, except for oxidative stress, where no effect was observed. By comparing U87^COX−2KO^ with U87^wt^, a significantly increased [^18^F]FMISO uptake was observed for COX-2 knockout cells after treatment with either S_HMC3, woFCS, or woGlucose (### *p* < 0.001) and with CoCl_2_ (# *p* < 0.05). In contrast, COX-2 knockout cells after co-culture with MG displayed a significant decrease in [^18^F]FMISO uptake after treatment with various compounds compared to U87^wt^ + MG (§ *p* < 0.05 for CoCl_2_, S_HMC3, and woGlucose; §§ *p* < 0.01 for H_2_O_2_).

### 2.2. In Vitro Experiments in Spheroids

COX-2 knockout spheroids exhibited significantly reduced growth compared to wildtype spheroids, stalling between days 5 and 19 (Figure 2a).

Spheroids from knockout cells showed a lower core cell density, as confirmed by H&E (Figure 2b) staining, and altered mechanical properties, revealed by Brillouin spectroscopy (Appendix A). Hypoxia regions, identified using pimonidazole staining, were more pronounced in COX-2 knockout spheroids, aligning with decreased Ki-67-positive cells (Figure 2b).

Spheroid glucose uptake ([^18^F]FDG) was significantly reduced in U87^COX−2KO^ compared to U87^wt^, correlating with lower viability and slower growth (*** *p* < 0.001) (Figure 2c). However, the hypoxia tracer uptake ([^18^F]FMISO) did not differ significantly between the spheroids from both cell types (Figure 2c). ROS levels in U87^COX−2KO^ spheroids significantly increased over time (### *p* < 0.001), contrasting with the decline observed in U87^wt^ spheroids (Figure 2d).

In U87^COX−2KO^ spheroids, a decreased expression of COX-1, the endothelial cell adhesion molecule E-selectin, the cytokine CXCL2, Fibronectin, and the monocyte/macrophage marker CD68 was found compared to U87^wt^ spheroids. CD31, an endothelial cell marker, remained unchanged (Figure 2e). We observed a slight increase in CD44 expression in the U87^wt^ spheroids over time. In contrast, the COX-2 knockout spheroids showed a slightly higher CD44 expression on day 5 and 12, which then was lower on day 19 compared to the wild-type spheroids. (Appendix A).

### 2.3. In Vivo Effects of COX-2 Knockout in Subcutaneous Xenograft Models

The growth rates of U87^COX−2KO^ tumors, compared to U87^wt^, were significantly slower in a subcutaneous xenograft model (*** *p* < 0.001) (Table 1 and Table 2; Figure 3a). The relative growth rate (change of tumor size in percent of the initial size per week), calculated from the reciprocal value of the doubling rate (days), of U87^COX−2KO^ was only 50% (0.9 ± 0.4) of U87^wt^ (2.0 ± 1.0).

Two distinctive patterns were observed when U87^COX−2KO^ cells were co-cultured with either bone marrow-derived MCs/Mϕs or MG cells prior to injection. The suppression of the growth rate was reversed when COX-2 knockout cells were co-cultured with bone marrow-derived MC (2.0 ± 0.7) or Mϕ (1.8 ± 0.7) cells (*** *p* < 0.001) (Figure 3b, right panel), This suggests a critical role for infiltrating immune cells in modulating tumor growth compared to U87^wt^ co-cultured with MCs or Mϕs (Figure 3b, left panel). In contrast, co-cultivation with microglia (MG) (0.4 ± 0.6) further enhanced the suppression of tumor growth in COX-2 knockout cells (** *p* < 0.01) (Figure 3c, right panel) compared to U87^wt^ cells (Figure 3c, left panel).

Subcutaneous COX-2 knockout tumors and multi-cellular tumors with MG displayed significantly reduced [^18^F]FMISO uptake compared to U87^wt^ and U87^wt^ + MG, respectively (*** *p* < 0.001) (Figure 3d, left panel). However, no difference in glucose ([^18^F]FDG) uptake was observed between the two groups (Figure 3d, right panel).

### 2.4. In Vivo Effects of COX-2 Knockout in Orthotopic Xenograft Models

Orthotopic xenografts confirmed prior findings of the subcutaneous xenograft model. COX-2 knockout tumors exhibited a significantly reduced volume compared to those from wildtype cells (* *p* < 0.05). In eight out of eight animals injected with U87^wt^, the average tumor volume was 3.72 mm^3^ at 27.75 days (Figure 4a). In five out of eight animals injected with U87^COX−2KO^, the average tumor volume was 1.51 mm^3^ at 67.2 days (Figure 4b).

H&E staining of U87^COX−2KO^ tumors revealed altered cellular morphology, with large, bubbled cells indicative of increased necrosis or apoptosis, which was not observed in U87^wt^ (Figure 5a).

Quantitative histology using HALO^®^ confirmed significantly reduced Ki-67-positive cells per mm^2^ in U87^COX−2KO^ tumors (** *p* < 0.01) (Figure 5b), despite no difference in overall cell density (Appendix A). Analysis of Ki-67 staining intensity (weak, moderate, and strong, based on identical thresholds across all sections) showed a significant genotype effect [F(1,24) = 45.47, *p*  <  0.0001], with reductions seen across all intensity levels. A post hoc test revealed that reductions in KI-67-positive cells were equally driven by all cells of U87^COX−2KO^ (*p* > 0.01 and *p* < 0.001), irrespective of the staining intensity (Appendix A).

Immunostaining for microglia (IBA1-positive) showed no difference in density within the tumor (Figure 5c) and an area of 150 µm around it (Appendix A). Albeit generally low across both tumor entities, the density of GFAP-positive astrocytes was significantly higher within U87^COX−2KO^ tumors. This effect was even more pronounced in the surrounding 150 µm area, where glial scar formation was prominent (*p* > 0.05; Figure 5d; Appendix A).

Segmentation validation with isotype controls confirmed negligible false positives for Ki-67 (≤ 2.28 cells/mm²) (Appendix A) and none for IBA1 (Appendix A). Percent area-based analysis replicated the findings for KI-67, IBA1, and GFAP (Appendix A), supporting the accuracy of our segmentation-based quantitative approach. A tendency towards decreased CD44 expression was found in tissue sections from U87^COX-2KO^ cells, orthotopically implanted cells, when compared sections of animals implanted with wildtype cells (Appendix A).

## 3. Discussion

Pathophysiological upregulation of COX-2 is strongly linked to GBM progression and poor prognosis. In contrast, expression in healthy tissue remains low, primarily localized to macrophages and microglia [9]. Within the glioma TME, both tumor cells and TAMs utilize the COX-2/PGE2 axis in autocrine and paracrine signaling [17,18]. This interaction is complex, involving multiple feedback loops, and it is unclear whether immune cells alter tumor growth or are influenced by it [3,19]. Unlike peripheral cancers, where the immune response can be suppressed via pathways like PD1/PD-L1 [20], GBM progression is shaped by the unique brain environment, including the blood–brain barrier and glial cell activity [21]. The enzyme COX-2 significantly impacts cell proliferation, invasion, metastasis, and immunosuppression, making it a promising therapeutic target in GBM [22]. The U87 glioblastoma model was chosen due to its wide application as a GBM model. Furthermore, it allows the interpretation of the obtained results in comparison with those already described for the model [23].

Additional inhibition of COX-2 in the TME was beyond the scope of this study. Instead, we focused on the effect of a complete COX-2 knockout with regard to tumor growth and its effect on the tumor microenvironment and vice versa. The CRISPR/Cas9-mediated COX-2 knockout in U87 glioblastoma cells significantly suppressed growth in spheroids and xenograft models but not in adherent cells. This suggests that COX-2 primarily impacts the interplay between tumor cells and the TME rather than intrinsic tumor cell proliferation. Measurable changes in adherent COX-2 knockout cells indicate G1 cell cycle arrest, increased intrinsic hypoxia, reactive oxygen species (ROS) formation, and mitochondrial activity, triggering or accelerating apoptotic and potentially autophagic pathways.

Previous studies have shown that the inhibition, silencing, or deletion of COX-2 can alter cell cycle progression, proliferation, and invasion in various tumor models [9,24,25,26]. In this study, we demonstrate that U87COX-2KO glioblastoma cells predominantly exhibit G1 cell cycle arrest without affecting in vitro proliferation. This arrest may persist until the cells receive signals to either resume proliferation or undergo cell death, aligning with similar findings in hepato-cellular carcinoma cells after COX-2 silencing [25]. Based on our data we assumed that the G1 arrest does not inevitably lead to apoptosis in vitro. Under optimal nutrient conditions, cells resume proliferation, and the G1 arrest is only transient but detectable. However, in nutrient-limited environments—such as spheroids or subcutaneous/orthotopic models—this balance collapses, resulting in reduced proliferation and induction of apoptosis.

It is important to note that COX-2 expression can vary considerably between cell lines of a tumor entity and, more generally, between cell lines of different tumor entities [27,28]. Influences on COX-2 expression, e.g., through experimental pharmacological interventions, are also characterized by high variability [29]. This makes direct comparisons between studies or generalizable conclusions difficult. As published previously, COX-2 inhibitors are able to reduce proliferation and colony formation in U87MG neurospheres in vitro [30]. However, we observed no reducing effects of COX-2 knockout on proliferation, colony formation, or motility in vitro, consistent with prior findings. We rather assume that off-targets effects of COX-2 inhibitors could be responsible [24,26]. Similarly, COX-2 knockout in A2058 melanoma cells did not affect proliferation but reduced invasion and was found not to release prostaglandin E2 (PGE2) [26]. Adherent U87^wt^ and U87^COX-2KO^ cells exhibited low basal levels of PGE2, which were unaffected by the COX-2 status. However, PGE2 levels significantly increased upon incubation with arachidonic acid. While PGE2 production during inflammation is predominantly mediated by the COX-2 pathway [31], we assume it can also occur via COX-1 and cytosolic prostaglandin E2 synthase in the central nervous system [32,33].

U87^COX−2KO^ cells show higher intrinsic hypoxia, ROS, and mitochondrial activity, which was consistent even under further stressors mimicking the TME. Normoxia and hypoxia in cell cultures differ significantly from tissue oxygen levels, with standard 20% O₂ incubators exceeding the physiological peri-cellular oxygen levels. Accurate hypoxia models should mimic in vivo oxygen levels when known [34]. The elevated [¹⁸F]FMISO uptake in adherent COX-2 knockout cells may result from self-inflicted hypoxia due to high oxygen consumption for ROS production or increased mitochondrial density [35], leading to inhibited growth and enhanced apoptosis in spheroids and in vivo models. It can be assumed that the upregulation of cytochrome c in U87^COX−2KO^ cells is associated with mitochondrial membrane breakdown due to increased ROS or apoptotic conditions involving Bcl-2 [36,37,38]. Cancer signaling pathway arrays showed significant upregulation of MAPK15 and p53, likely driven by elevated ROS, promoting apoptosis and autophagy [39,40]. This aligns with reports of COX-2 inhibitors affecting survival and apoptosis pathways [41].

In spheroids, which better mimic in situ tumors with organized proliferative and necrotic regions, COX-2 knockout significantly impaired growth and proliferation. Similar findings were reported for melanoma cells and prostate cancer, where COX-2 or mPGES-1 inhibition reduced PGE2 production and spheroid growth [26,42]. COX-2 knockout also downregulated proteins involved in tumor–macrophage crosstalk, including CXCL2 and E-selectin, with CXCL2 being a pro-angiogenic chemokine linked to glioma progression and a potential therapeutic target [43,44,45]. The altered metabolism and increased ROS in U87^COX−2KO^ cells may activate anti-tumor pathways via apoptosis, contributing to impaired tumor growth. The discrepancies between the in vitro - and spheroid/in vivo data may be due to COX-2 influencing tumor growth indirectly via the microenvironment rather than directly affecting cancer cell proliferation. Under optimal in vitro conditions, COX-2 knockout effects can likely be compensated. This balance is disrupted under more complex, tumor-like conditions in spheroids and in vivo, revealing the inhibitory impact. In vitro, cells may remain in a survival mode that cannot be sustained in the presence of microenvironmental stress.

Consistent with the spheroid findings, U87^COX−2KO^ cells exhibited suppressed proliferation and tumor growth in both subcutaneous and orthotopic xenograft models. This aligns well with previous melanoma studies [26] and reports on COX-2 knockdown in other cell lines [24,25,46]. To investigate the tumor-associated macrophage (TAM)–glioma cell crosstalk, multi-cellular tumors with human monocytes (MCs), macrophages (Mϕs), or microglia (MG) were implanted. Peripheral monocytes and macrophages tended to accelerate U87^wt^ tumor growth rates. More importantly, their presence abolished the growth suppression observed for U87^COX−2KO^, while microglia further it. These findings highlight TAMs’ crucial role in glioma development and progression. Similar effects have been observed in vitro, where microglia co-cultures affected glioblastoma cell proliferation and migration, and even induced drug resistance at low ratios [47]. This underscores the complexity of TAM–tumor interactions in driving glioblastoma progression.

Interestingly, CD44, a known GBM stem cell marker [48], showed a trend towards reduced—though not significant—expression in orthotopic xenografts after COX-2 knockout. In COX-2 knockout spheroids, CD44 reduction was only observed on day 19. This, in part, is consistent with data published by Lombardi et al., who found that COX-2 upregulation (via temozolomide) promoted an immunosuppressive microenvironment with increased CD44 expression. In turn, COX-2 inhibition via celecoxib reduced CD44 expression and counteracted this effect, which was reversed by exogenous prostaglandin E2, underscoring the role of COX-2 or respective prostaglandins [18].

In an orthotopic glioblastoma model, a trend towards increased numbers of IBA-positive microglia with amoeboid morphology was observed at tumor margins, consistent with prior findings [49]. However, COX-2 knockout did not affect IBA-positive cell density within the tumor margins or stroma, even though their overall morphology was drastically different from the resting microglia in both U87 entities. In contrast, GFAP-positive astrocytes were moderately increased in and around U87^COX−2KO^ tumors, potentially due to more efficient glial scar formation or autophagy, as the GFAP has been implicated in mitochondrial transfer and autophagy pathways via p38/MAPK and mTOR downregulation [50,51]. COX-2 expression and PGE2 secretion were previously shown to be upregulated in U87 cells co-cultured with M2 macrophages, promoting vascularization through COX-2 activation and highlighting crosstalk between inflammatory and perivascular microenvironments [52]. Similarly, phorbol ester-treated THP-1 cells enhanced cancer cell invasion and angiogenesis via COX-2-dependent MMP-9, VEGF-A, and bFGF release [53]. The factors driving TAM–glioma cell crosstalk remain unclear, but TAMs and tumor cells dynamically alter their expression profiles based on environmental signals [19]. Array data suggest COX-2 knockout downregulates proteins essential for TAM interaction, while the upregulation of p53 and MAPK15 may promote apoptosis and autophagy in knockout cells [54].

## 4. Materials and Methods

### 4.1. Cell Culture Conditions

The human glioblastoma cell line U87 MG (# HTB-14) and human microglia cell line HMC3 (CRL-3304; referred to as MG throughout the text) were purchased from ATCC. The human acute monocytic leukemia cell line THP-1 (ACC-16) was purchased from DSMZ. THP-1 monocytes (MCs) were differentiated into M0 macrophages (Mϕs) using 64 nM PMA (Phorbol 12-myristate 13-acetate) for three days.

All cells were cultured in Dulbecco’s modified Eagle’s medium (DMEM) supplemented with 10% (*v*/*v*) fetal calf serum and 1 U/mL penicillin/streptomycin (all reagents from Biochrom, Berlin, Germany) under normoxic conditions (37 °C, 5% CO_2_). For hypoxia experiments (extrinsic hypoxia model), cells were cultured at a reduced oxygen level (<1% *v*/*v*) in a special incubator equipped with an oxygen sensor (Gasboy; Labotect, Göttingen, Germany). To generate spheroids, U87^wt^ and U87^COX−2KO^ cells were harvested by trypsinization and seeded in 96-well non-adherent plates (Greiner Bio One, Frickenhausen, Germany) at a density of 2000 cells per well [in 100 μL of a 0.5% methylcellulose (Bio-Techne, Wiesbaden, Germany) and culture medium]. Spheroids were maintained for 5, 12, and 19 days.

### 4.2. Knockdown of COX-2 with CRISPR/Cas9

A CRISPR/Cas9 knockdown kit targeting human COX-2 was obtained from Santa Cruz Biotechnology (Dallas, TX, USA). U87 cells (2 × 10^5^) were seeded into six-well plates 24 h prior to transfection. Cells were transfected with 1 µg COX-2 CRISPR/Cas9 KO Plasmid and 1 µg COX-2 HDR Plasmid using UltraCruz^®^ Transfection Reagent (Dallas, TX, USA) in a serum-free growth medium, following the manufacturer’s protocol.

After 24 h, the medium was replaced with a complete growth medium containing 0.5 µg/mL puromycin for selection. Transfected cells were monitored via RFP fluorescence microscopy, and RFP-positive and puromycin-resistant cells were collected for monoclonal expansion. COX-2 knockout was confirmed by fluorescent microscopy and Western blotting.

### 4.3. Antibody Array

Cancer Biomarker (247 targets) and Cancer Signaling Phospho (269 targets) antibody arrays (tebu-bio, Offenbach, Germany) were used following the manufacturer’s protocol. Array images were assessed using tebu-bio. Mean values and statistical analysis was provided by the manufacturer. Changes were considered significant if the expression ratio between U87^wt^ and U87^COX-2KO^ was ≤ 0.5 (decrease) or ≥ 2.0 (increase). A value of 0.5 indicates that the protein amount has decreased by 50%, and a value of 2 means the protein amount has doubled.

### 4.4. Growth Rate Analysis

Cell growth was assessed using the IncuCyte^®^ Live-Cell Analysis System (Sartorius, Göttingen, Germany). Cells were seeded in 96-well plates at ~10% confluence and imaged every 2 h for 9 days. Growth rates were determined using the system’s integrated confluence algorithm.

### 4.5. RFP Control and Cell Cycle Analysis by FACS

RFP fluorescence in U87^COX−2KO^ was verified before each experiment using an Attune NxT CytKick flow cytometer (Thermo Fisher Scientific, Dreieich, Germany). For cell cycle analysis, ethanol-fixed cells were treated with 50 µg/mL RNase A and stained with 10 µg/mL propidium iodide before analysis by flow cytometry. The results were analyzed by FlowJo 7.6.1 software (FlowJo, LLC, Ashland, OR, USA).

### 4.6. Western Blot Analysis

SDS-PAGE and Western blotting were performed, as described previously [16]. PVDF membranes were incubated with anti-COX-2 (D5H5) XP^®^ (#12282; 1:400, Cell Signaling, Danvers, MA, USA); anti-COX1 (ab227513; 1:500, Abcam, UK); Cytochrome c (#MA5-15078; 1:400; Invitrogen, USA); anti-Phospho-MAPK15 (#PA5-99135, 1:500, Invitrogen, USA); anti-LOX (F-8) (sc-373995, 1:500, Santa Cruz Biotechnology, INC.; Dallas, Tx, USA); anti-GAPDH (G8795, 1:1000; Sigma-Aldrich, Darmstadt, Germany); and anti-β-actin (A5060; 1:1000, Sigma-Aldrich, Germany) antibodies, followed by incubation with an appropriate peroxidase-coupled secondary antibody (anti-rabbit IgG, A0545, Sigma-Aldrich, 1:5000; anti-mouse IgG, A9044, Sigma-Aldrich, 1:10.000). Bands were visualized using SuperSignal^®^ West Pico and Femto chemiluminescent Substrate (Thermo Fisher Scientific, Dreieich, Germany) and imaged with an MF-ChemiBIS Bio-Imaging System (Biostep GmbH, Burkhardtsdorf, Germany). Western blot densitometry and relative intensity of protein bands calculated using Aida 5.10 software. 

### 4.7. Histological Analysis of U87^wt^ and U87^COX−2KO^ Spheroids

On days 5, 12, and 19, the spheroids were fixed with phosphate-buffered paraformaldehyde (4%) for 30 min and processed in an alcohol series ending with isopropanol, followed by paraffin embedding. For the histological examination, 2 µm thick sections were mounted on SuperFrost Plus slides. Sections were deparaffinized, rehydrated, and stained with hematoxylin and eosin (H&E).

### 4.8. Spheroid Histology and Immunohistochemistry

The spheroids were fixed in 4% paraformaldehyde, embedded in paraffin, and sectioned (2–5 µm). Hematoxylin and eosin (H&E) staining was performed for structural analysis. For hypoxia detection, the spheroids were treated with 200 µM pimonidazole for 2 h before fixation, and subsequent immunohistochemical staining was performed for Ki-67 (Abcam, Cambridge, UK, #15580, 1:2000), CXCL2 (GRO-β) (Bio-Rad, Dreieich, Germany, #150439, 1:500), CD31 (Abcam #28346, 1:75), CD68 (Abcam, #125212, 1:500), Fibronectin (Abcam, #2413, 1:250), CD62E (E-selectin) (Abcam #18981, 1:500), and COX-1 (Abcam #109025, 1/:50). Stained slides were imaged using an AxioImager A1 microscope (Zeiss, Jena, Germany).

### 4.9. Oxidative Stress and Metabolic Assay

Reactive oxygen species (ROS) were measured using CellROX^®^ DeepRed (Thermo Fisher Scientific). Mitochondrial activity was assessed using MitoTracker^®^ Green FM dye. Lipid peroxidation was quantified by a thiobarbituric acid reactive substances (TBARS) assay (Cell Biolabs, San Diego, CA, USA).

For this purpose, U87^wt^ and U87^COX−2KO^ cells were treated with different agents for 24 h: H_2_O_2_ (200 μM) as a model of an oxidative stressor; CoCl_2_ (200 nM) as a hypoxia-inducing agent; supernatants of HMC3 microglia cells (S_HMC3); media without serum (woFCS); and media without glucose (woGlucose).

### 4.10. Radiotracer Uptake Assay

Glucose ([^18^F]FDG) and hypoxia ([^18^F]FMISO) were assessed in cells and spheroids. Cells were incubated with 0.5 MBq [^18^F]FDG for 1 h or 0.5 MBq [^18^F]FMISO for 4 h, washed, lysed, and radioactivity measured using a gamma counter. The protein-normalized uptake was reported as the percent initial dose (ID) per mg protein.

### 4.11. Subcutaneous Glioblastoma Xenografts

Animal experiments were conducted in accordance with German Animal Welfare Regulations and approved by the Landesdirektion Dresden Animal Ethics Committee (reference number DD24.1-5131/449/49). Female NMRI nude mice (Janvier Labs) were subcutaneously injected with 5 × 10⁶ U87^wt^, U87^COX−2KO^, or Mϕ cells, or 1 × 10⁶ MC or MG cells, each suspended in 100 µL PBS/Matrigel (50:50). Tumor size was measured three times per week, and volume was calculated as V = π/6 × (tumor length × tumor width^2^). The growth rate and doubling time were determined from exponential curve fitting. Non-growing tumors were assigned a growth rate and doubling time of zero at the time of the last measurement. Statistical comparisons were performed using the Mann–Whitney U test (*p* < 0.05) performed with R (v4.3.1).

### 4.12. Small Animal PET/CT Imaging of Subcutaneous Glioblastoma Xenografts

Small animal positron emission tomography (PET) was performed using a nanoPET/CT scanner (Mediso Medical Imaging Systems, Budapest, Hungary) with a field of view (FOV) of 9.6 × 10 cm, enabling the whole-body imaging of mice. U87^wt^ and U87^COX−2KO^ tumor-bearing mice were anesthetized with 9% desflurane (Suprane, Baxter, Heidelberg, Germany) in 30% oxygen/air (1 L/min) and intravenously injected with 8–12 MBq (228–342 MBq/kg) of [¹⁸F]FMISO or [¹⁸F]FDG in 0.2 mL of saline via a tail vein catheter. Imaging was performed dynamically for 60 min ([¹⁸F]FDG) or 30 min ([¹⁸F]FMISO, 240 min post-injection).

CT scans were recorded for anatomical referencing and attenuation correction. Data were reconstructed using the Tera-Tomo™ 3D algorithm and analyzed with ROVER software v3.0.73h (ABX, Radeberg, Germany), generating maximum intensity projections (MIPs). Tumors were delineated to create 3D volumes of interest (VOIs). Standardized uptake values (SUVmean and SUVmax) were calculated for VOIs and analyzed using GraphPrism9 Software, San Diego, CA, USA).

### 4.13. Orthotopic Glioblastoma Xenograft Tumor Model

Female athymic nude mice (Rj:NMRI-Foxn1 nu/nu) were used for tumor implantation at the age of 8 weeks (25–30 g), as outlined in Figure 6b. During microsurgery, mice were anesthetized with a mixture of oxygen/air and desflurane (9–12% depending on respiratory rates) under aseptic conditions and gentle heating. Animals were placed into a Stoelting stereotactic frame (just for mouseTM, Stoelting Europe, Dublin, Ireland). A midline incision was performed, and a burr hole drilled, 0.5 mm anterior and 2.4 mm lateral to the bregma. About 1.6–8 × 10^4^ U87^wt^ or U87^COX−2KO^ cells, respectively, were suspended in Hank’s Buffered Salt Solution (HBSS, 1×). This suspension was injected at 3.0 mm depth into the brain parenchyma with a flow of 0.1 μL/min using a 10 μL Hamilton syringe (Figure 6a). After injection, the burr hole was closed with bonewax (Ethicon, Cincinnati, OH, USA), the scalp incision was sutured (Vicryl 6.0, Ethicon), and the surface was antiseptically cleaned.

### 4.14. MRI Imaging

Tumor volumes were measured using a dedicated 7T small animal magnetic resonance imaging scanner (MRI, ParaVision software 6.0.1., Bruker, Ettlingen, Germany), as outlined in Figure 6b. For MRI measurements, a T2-weighted measuring sequence (TRARE) was applied with an echo and repetition time of 35 ms and 2500 ms, respectively. The spatial resolution was set to 78 µm in an xy direction. The slice thickness was set to 0.7 mm. Hereinafter, tumor volumes were quantified using the software ROVER (version v3.0.73h; ABX GmbH, Radeberg, Germany). To determine the tumor size, all voxels of the tumor in the slices were manually selected, and the volume of the generated ROI was calculated.

### 4.15. Immunohistochemistry of U87^wt^ and U87^COX−2KO^ Brain Tumors

The experimental pipeline of tissue cutting, staining, and analysis is shown in Figure 6c,d. Brain tumors were collected from mice with MRI-confirmed growth near the endpoints. After deep anesthesia and cervical dislocation, animals were perfused, and their brains were extracted. They were further fixed in 4% PFA for 24 h and stored in PBS with 0.1% sodium azide and 20% sucrose for 3 days. The brains were paraffin-embedded, and 5 µm coronal sections were prepared, with 3–4 sections per slide and a spacing of 75–500 µm using a microtome (Microm HM340E; Thermo Fisher Scientific, Cambridge, UK).

The sections were first deparaffinized and rehydrated. For hematoxylin and eosin (H&E) staining [27], the sections were immersed in Mayer’s hematoxylin for three minutes, followed by blueing, counterstaining with eosin for 30 s, and dehydration through an ethanol gradient. Finally, slides were cleared with RotiClear^®^ and mounted using RotiMount^®^ (Carl Roth, Karlsruhe, Germany) For immunohistochemistry, antigen retrieval was performed by heating sections in a 10 mM citrate buffer at 95 °C for 20 min. Endogenous peroxidase activity was quenched with 3% hydrogen peroxide, and nonspecific binding was blocked with 10% fetal calf serum in TBS–Tween for one hour at room temperature. The sections were incubated overnight at 4 °C with primary antibodies targeting the glial fibrillary acidic protein (GFAP, Agilent, Santa Clara, CA, USA) #Z0334, 1:4000 of 3.9 mg/mL), IBA1 (Wako Pure, Neuss, Germany, 019-19741, 1:1000 of 0.5 mg/mL), and Ki67 (Abcam, Cambridge, UK, #15580, 1:1000 of 0.8 mg/mL). After washing, biotinylated anti-rabbit secondary antibodies were applied, followed by incubation with ExtrAvidin peroxidase. Visualization was achieved using an AEC substrate, and sections were mounted with a gelatin-coating solution. Isotype controls were included using rabbit IgG at equivalent concentrations to the primary antibodies.

### 4.16. Data Acquisition and Analysis

All slides were scanned using an Axioscan Z1 (Zeiss, Germany) with a 20× objective (EC-Plan Neofluar 20×/0.5) and a color camera (Hitachi HV-F202SCL), yielding a scaling of 0.221 µm × 0.221 µm per pixel.

Quantitative analysis was performed using HALO^®^ 3.3, with a region of interest (ROI) covering the extent of the tumor (KI67, IBA1, and GFAP). Additionally, the stroma was analyzed for IBA1 and GFAP, using a function of HALO^®^ to define a new area (150 µm) around the tumor’s ROI. Empty areas (e.g., from ventricles or defects) and artifacts (blushes or folds) were excluded from the analysis through HALO’s built-in exclusion tool.

The primary analysis for Ki-67 was performed using the CytoNuclear module, with outcome measures being total and Ki-67-positive cells per mm^2^, along with Ki-67 staining intensity (weak, moderate, and strong). An additional control involved using the Area quantification module to verify segmentation results.

IBA1 and GFAP analysis was performed using the HALO^®^ Microglia module. Here, negative and immune-positive cells per mm^2^ were the primary outcome measures. As glial morphology within the tumor was changed substantially for IBA1, a secondary analysis (see Appendix A) using the Area quantification module was performed for IBA1 and GFAP staining, confirming findings from the segmentation algorithm.

All quantitative analysis settings were tuned and tested on multiple images. For the subsequent analysis, a single setting was employed for all images. Data from one slide was averaged, and the mean was calculated across the different anatomical planes.

### 4.17. Statistical Analysis

Data are presented as mean ± SD or SEM from at least three independent experiments. Statistical analyses were performed using GraphPad Prism 9. To compare the effects of U87^wt^ or U87^COX−2KO^ implantation, a two-tailed t-test was used after checking for the normality of the data (Kolmogorov–Smirnov). The relationship between staining intensity and COX-2 phenotype was investigated using a two-way ANOVA with Šídák’s multiple comparisons test, if appropriate. For all analyses, a value of *p* < 0.05 was considered statistically significant (* *p* < 0.05, ** *p* < 0.01, and *** *p* < 0.001).

## 5. Conclusions

In conclusion, these findings highlight the potential of complete COX-2 knockout as a therapeutic strategy in GBM, particularly when combined with targeting peripheral macrophages and maintaining a stable microglia population to enhance apoptotic cell clearance and suppress glioblastoma progression. These multi-target therapies, including tissue-specific CRISPR/Cas9 genome editing, hold significant promise in addressing GBM heterogeneity [59].

## Figures and Tables

**Figure 1 ijms-26-06198-f001:**
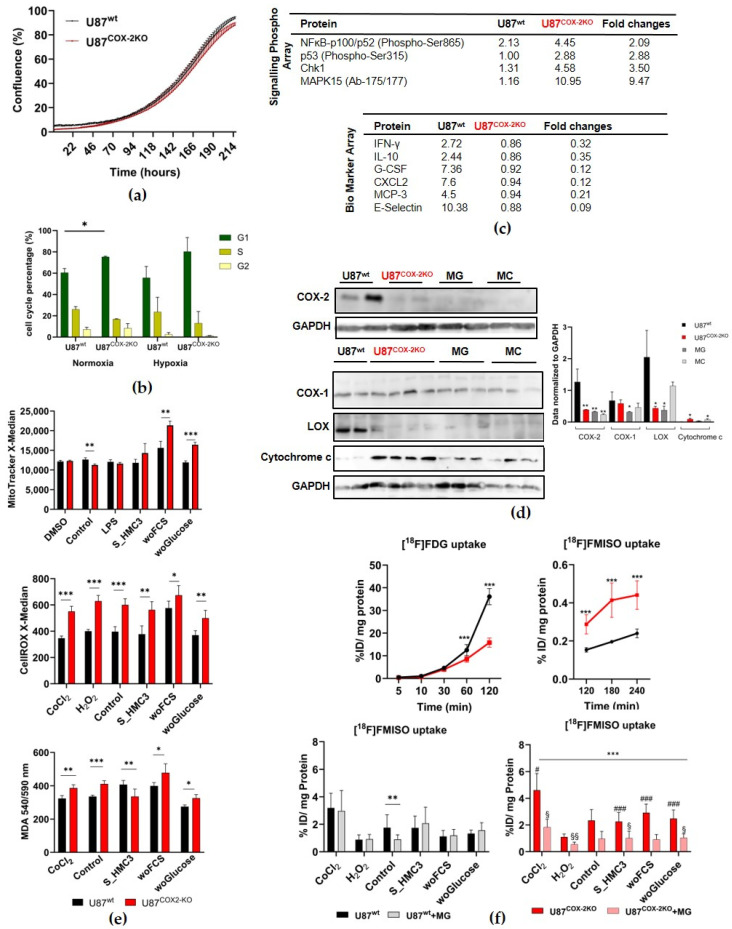
A comparative analysis of COX-2 knockout (red) and wildtype (black) U87 monolayer cell cultures. (**a**) Analysis of cellular growth of U87^wt^ and U87^COX−2KO^ over 9 days (*n* = 3). (**b**) Cell cycle analysis under normoxic and hypoxic conditions using flow cytometry after staining with propidium iodide. (**c**) One sample analysis with six replicates of a “Signaling Phosphor Array” and a “Cancer Biomarker Antibody Array”. Mean values and statistical analysis was provided by the manufacturer. Changes were considered significant if the expression ratio between U87^wt^ and U87^COX-2KO^ was ≤0.5 (decrease) or ≥2.0 (increase). (**d**) Immunoblot analysis using indicated antibodies. Western blots were performed either in a duplicate (upper panels) or a multiple (lower panels) and normalized to GAPDH. (**e**) Determination of mitochondria (MitoTracker green via flow cytometry), ROS (CellROX via flow cytometry), and lipid peroxidation (MDA via TBARS Assay) after different stressor treatment for 24h: H_2_O_2_ (200 μM) as a model of an oxidative stressor; CoCl_2_ (200 nM) as a hypoxia-inducing agent; supernatants of HMC3 microglia cells (S_HMC3); media without serum (woFCS); media without Glucose (woGlucose). (**f**) Uptake of [^18^F]FDG over 120 min, and [^18^F]FMISO over 240 min (upper panel) in U87^wt^ (black) and U87^COX-2KO^ (red) cells. Additional uptake of [^18^F]FMISO after coculture with MG cells (1:1) and different treatments for 24 h. Results of three independent experiments with quadruple determination. Values in the graphs represent the mean ± SEM. * *p* < 0.05, ** *p* < 0.01, *** *p* < 0.001. For (**d**) * *p* < 0.05, ** *p* < 0.01 vs. U87^wt^. For (**f**) ** *p* < 0.01, *** *p* < 0.001 vs. control; # *p* < 0.05, ### *p* < 0.001 vs. treatment; § *p* < 0.05, §§ *p* < 0.01 U87^wt^ + MG vs. U87^COX-2KO^ + MG.

**Figure 2 ijms-26-06198-f002:**
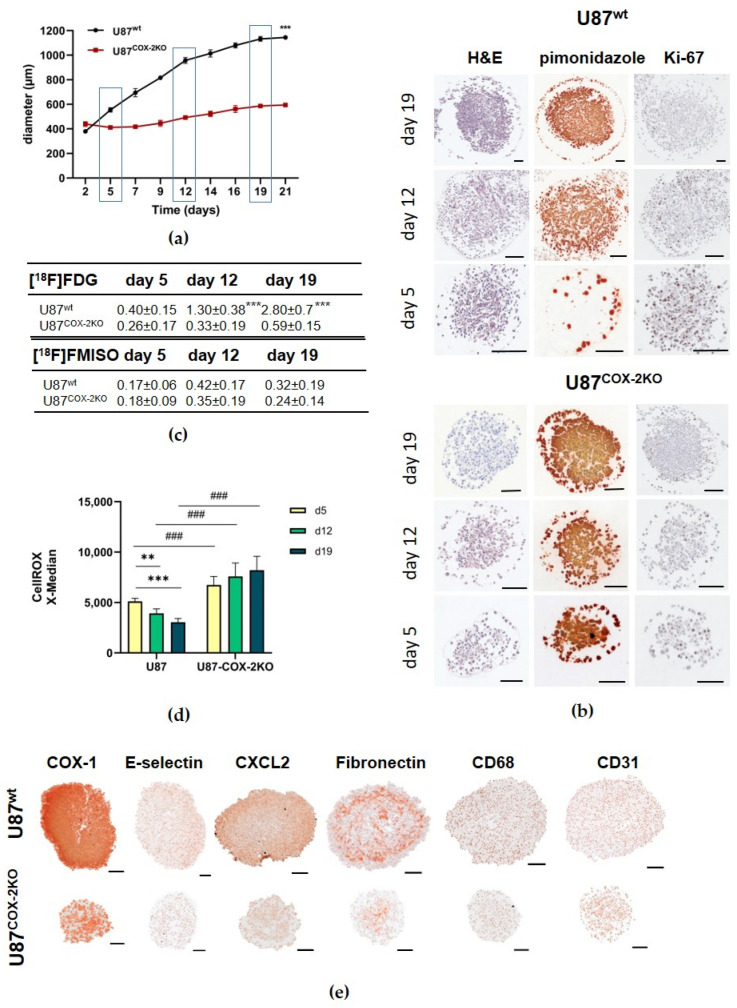
Comparative analysis of COX-2 knockout (red) to wildtype (black) U87 spheroids. (**a**) Growth curves of U87^wt^ and U87^COX−2KO^ over 21 days (*n* = 12). (**b**) Representative images of spheroids after days 5, 12, and 19 with staining for H&E, pimonidazole, and Ki-67. (**c**) Uptake of [^18^F]FDG over 120 min and [^18^F]FMISO over 240 min in %ID/g. (**d**) Determination of ROS activity (CellROX) in U87^wt^ and U87^COX−2KO^ spheroids after 5, 12, and 19 days. Results of two independent experiments with quadruple determination. (**e**) Immunostaining of tissue sections from U87^wt^ and U87^COX-2KO^ spheroids, using indicated antibodies. Values in the graphs (**a**,**c**,**d**) represent the mean ± SEM. *** *p* < 0.001. For (**d**) ** *p* < 0.01, *** *p* < 0.001 vs. days; ### *p* < 0.001 U87^wt^ vs. U87^COX-2KO^. Scale bar: 100 µm.

**Figure 3 ijms-26-06198-f003:**
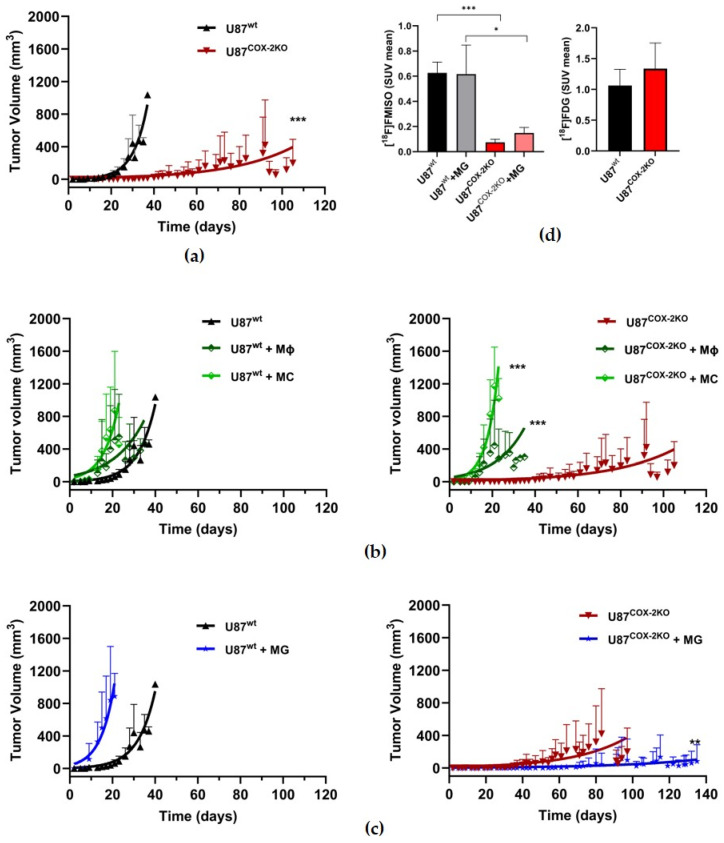
Subcutaneous xenograft growth of U87^wt^ (*n* = 44) and U87^COX−2KO^ (*n* = 38) in NMRI nude mice. (**a**) Comparison of tumor volume over time between U87^wt^ (black) and U87^COX−2KO^ (red). (**b**) Comparison of tumor volume over time between U87^wt^ (black) with either U87^wt^ + Mϕs (dark green, *n* = 23) or U87^wt^ + MCs (green, *n* = 18) (left panel). Comparison of tumor volume over time between U87^COX−2KO^ (red) with either U87^COX−2KO^ + Mϕs (dark green, *n* = 23) or U87^COX−2KO^ + MCs (green, *n* = 18) (right panel). (**c**) Comparison of tumor volume over time between U87^wt^ (black) and U87^wt^ + MG (blue, *n* = 13) (left panel). Comparison of tumor volume over time between U87^COX−2KO^ (red) and U87^COX−2KO^ + MG (blue, *n* = 23) (right panel). (**d**) Radiotracer uptake 240 min after i.v. injection of 8–12 MBq [^18^F]FMISO in xenografts from U87^wt^ and U87^COX−2KO^ cells or as multi-cellular tumors with MG cells (left panel). Radiotracer uptake 60 min after i.v. injection of 10 MBq [^18^F]FDG in U87^wt^ and U87^COX−2KO^ (right panel), respectively, in NMRI nude mice xenografts. (SUV: standardized uptake value.) Values in the graphs represent the mean ± SEM. * *p* < 0.05, ** *p* < 0.01, *** *p* <0.001.

**Figure 4 ijms-26-06198-f004:**
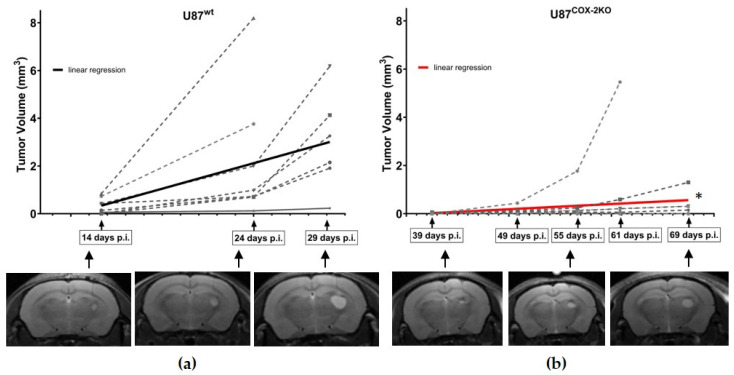
Orthotopic xenograft growth of U87^wt^ and U87C^OX−2KO^ in NMRI nude mice. In vivo T2-weighted MRI images and calculated tumor volumes at various timepoints of (**a**) U87^wt^ implanted tumor cells in 8 out of 8 animals (dotted lines) and (**b**) U87^COX−2KO^ implanted tumor cells in 5 out of 8 animals (dotted lines), along with corresponding representative images. Value in the graph represents the mean ± SEM. * *p* < 0.05.

**Figure 5 ijms-26-06198-f005:**
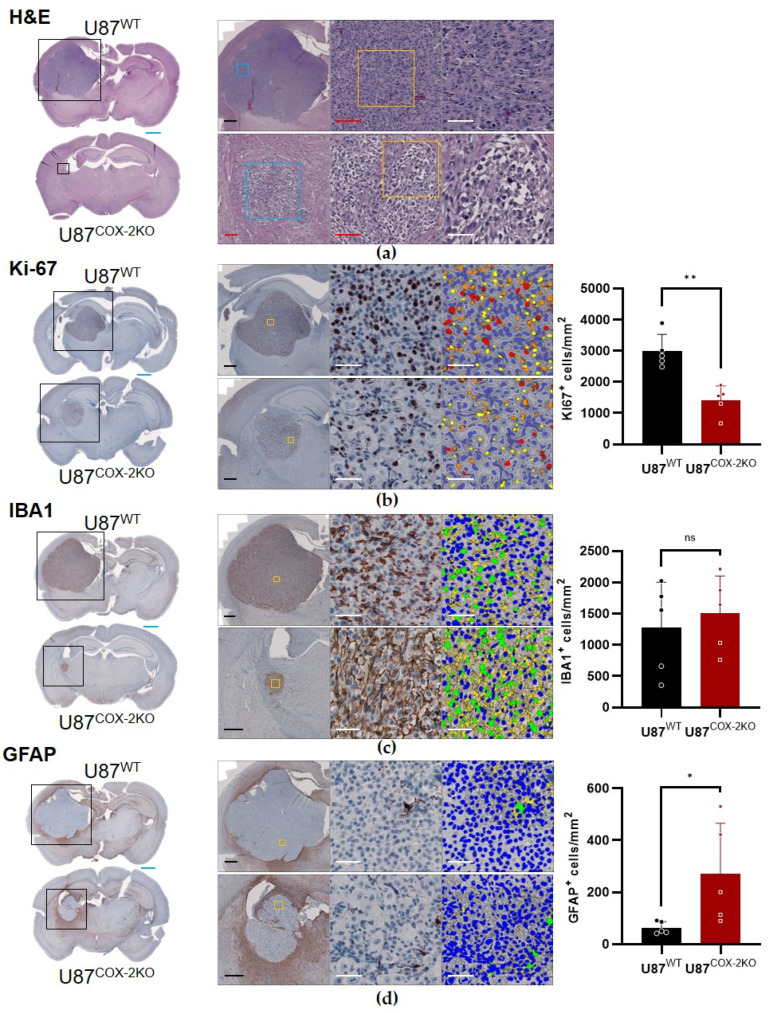
The qualitative (H&E) and quantitative histology of Ki-67, IBA1-, and GFAP-positive cells in the tumor. Representative whole brain photomicrographs for H&E (**a**), Ki-67 (**b**), IBA1 (**c**), and GFAP (**d**) staining are presented in the left panel (upper micrograph for U87^wt^ and lower for U87^COX-KO 2KO^, respectively). A black rectangle indicates the magnified area shown in the middle left panel. A further magnification (yellow rectangle) and its location are provided in the middle right panel. For H&E, the right panel shows a second (blue rectangle) and a third magnification (yellow rectangle). For all other stains, the right panel shows the segmentation results as a color-coded overlay. For Ki-67 (B, right panel), this indicates hematoxylin-positive (blue) and Ki-67-positive cells, color-coded for staining intensity (yellow: weak; orange: moderate; and red: strong). For IBA1 and GFAP (C/D, right panel), this indicates hematoxylin-positive (blue) and IBA1/GFAP-positive cells (green). The processes are labeled yellow (quantified but not analyzed). Graphs show the mean ± SD of stain-positive cells per animal (from 6–12 sections) of U87^wt^ and U87^COX2-KO^ (n = 5 each). Scale bars are color-coded with light blue: 1000 µm (whole brain photomicrographs); solid black: 500 µm; red: 100 µm and white: 50 µm (all magnified views). * *p* < 0.05, ** *p* < 0.01 U87^wt^ vs. U87^COX2-KO^.

**Figure 6 ijms-26-06198-f006:**
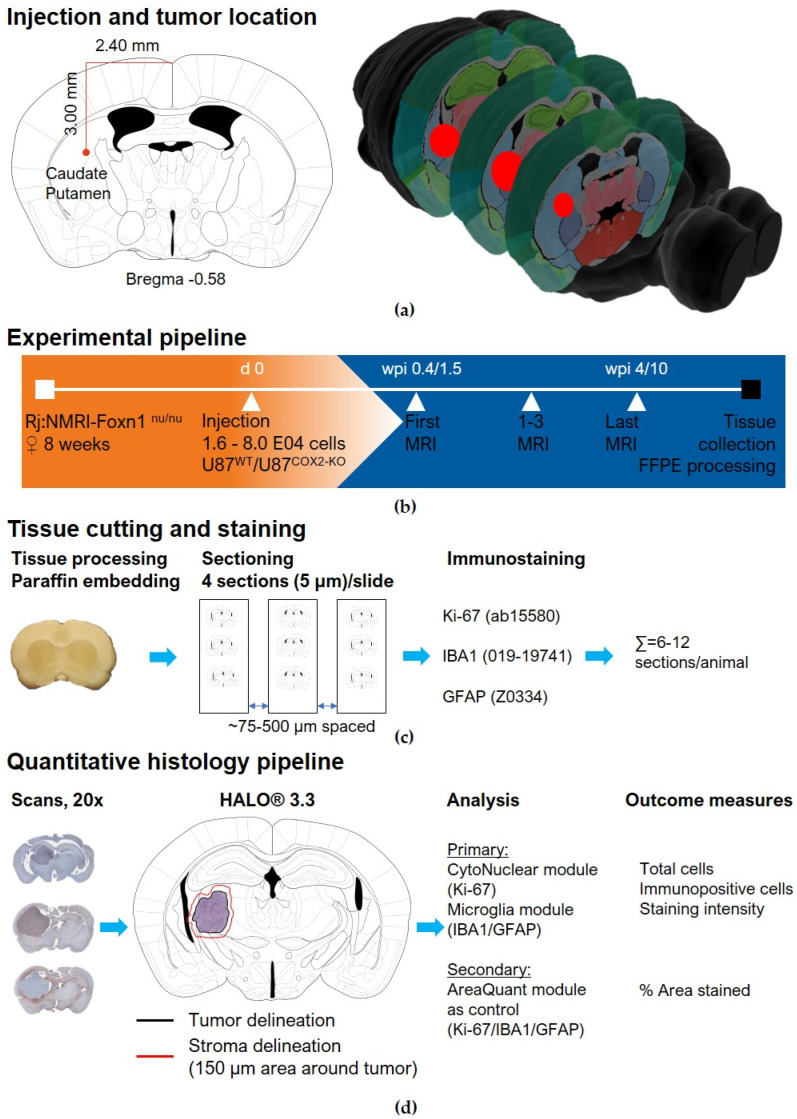
Overview of experimental details and protocol for in vivo experiments and downstream quantitative histology. (**a**) Brain region chosen for injection with coordinates and schematic of approximate tumor size. (**b**) Experimental pipeline for animal experiments. (**c**) Tissue processing, sectioning, and immunostaining procedures. (**d**) Quantitative histology workflow from imaging, region of interest creation, and analysis type to outcome measures used for statistical comparison. Figure a was created using material from [55] and the “Blue Brain Cell Atlas” (URL accessed 12/11/2024) based on publications by [56,57,58].

**Table 1 ijms-26-06198-t001:** Relative growth rate for xenografts from U87^wt^ and U87^COX-2KO^ cells and multi-cellular tumors with either Mϕ, MC, or MG.

	N	Mean ± SD	Median
U87^wt^	44	2.0 ± 1.0	1.8
U87^wt^ + MC	18	2.3 ± 1.1	1.7
U87^wt^ + Mϕ	23	2.1 ± 1.1	1.8
U87^wt^ + MG	13	3.1 ± 0.4	3.0
U87^COX-2KO^	38	0.9 ± 0.4	1.0
U87^COX-2KO+MC^	18	2.0 ± 0.7	1.8
U87^COX-2KO +Mϕ^	23	1.8 ± 0.7	1.8
U87^COX-2KO + MG^	23	0.4 ± 0.6	0.0

**Table 2 ijms-26-06198-t002:** Relative growth rate along with statistics for xenografts from U87^wt^ and U87^COX-2KO^ cells and multi-cellular tumors with either Mϕ, MC, or MG.

Parameters	*p* Value
U87^wt^ vs U87^COX-2KO^	<0.001
U87^wt^ vs U87^wt^ + MC	0.57
U87^wt^ vs U87^wt^ + Mϕ	0.98
U87^wt^ vs U87^wt^ + MG	0.3
U87^COX-2KO^ vs U87^COX-2KO^ + MC	<0.001
U87^COX-2KO^ vs U87^COX-2KO^ + Mϕ	<0.001
U87^COX-2KO^ vs U87^COX-2KO^ + MG	<0.002
U87^wt^ + MC vs U87^COX-2KO^ + MC	0.78
U87^wt^ + Mϕ vs U87^COX-2KO^ + Mϕ	0.67
U87^wt^ + MG vs U87^COX-2KO^ + MG	<0.001

Analyses were performed with two-sided Wilcoxon test.

## Data Availability

The datasets used and/or analyzed during the current study are available from the corresponding author upon reasonable request.

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
