# Peer review of "Exploring the Role of Peripheral Macrophages in Glioma Progression: The Metabolic Significance of Cyclooxygenase-2 (COX-2)"

_ijms, 2025, doi:10.3390/ijms26136198_

Round 1
Reviewer 1 Report
Comments and Suggestions for Authors
Pietzsch et al. describe the impact of COX-2 knock-out by engineering U87MG cells using CRISPR/Cas9 technology. The authors use a plethora of different methods and preclinical model systems to comprehensively decipher differences between COX-2 sufficient and knock-out U87MG cells.
According to the presented data monolayer cultures of U87MG and subcutaneous xenografts resulted in less convincing data as compared to spheroid and orthotopic PDX models of the investigated U87 clones. Thus, I suggest the following additional experiments.
- Spheroid experiments as presented in Figure 2 would benefit from quantification analyses (e.g using HALO software).
- The reduced PDX growth in U87COX-2KO cells is convincing and underlined by Ki67 staining and quantification. Nevertheless, it appears that COX-2 KO cells undergo cellular plasticity, indicated by a more spindle-shaped cell morphology. The authors describe an increased invasive capacity upon COX-2 KO, thus I suggest to investigate cellular invasiveness in PDX models by immunohistochemistry. In this respect, CD44 was already discussed as marker in mesenchymal glioma subpopulations. Staining of CD44 in orthotopic PDX models and spheroids would support this data.
- As the submitted study is solely based on one cell line, I suggest to demonstrate the effects of COX-2 on glioblastoma aggressiveness (hypoxia, proliferation, ROS induction etc.) by including more cell models. In case the establishment of additional CRISPR-KO-models is not feasible, this work would also benefit from treatment experiments with COX-2 inhibitors e.g Celecoxib in additional glioblastoma cells (spheroid assays). This data would further strengthen a clinical translation of COX-2 inhibition in glioblastoma.
Author Response
Comments 1: Spheroid experiments as presented in Figure 2 would benefit from quantification analyses (e.g using HALO software).
Response 1: Thank you for pointing this out. Unfortunately, we were only granted temporary guest access to the HALO software. Due to time constraints, we were forced to limit our analysis of the immunohisto-chemical staining to the orthotopic model.
Comments 2: The reduced PDX growth in U87COX-2KO cells is convincing and underlined by Ki67 staining and quantification. Nevertheless, it appears that COX-2 KO cells undergo cellular plasticity, indicated by a more spindle-shaped cell morphology. The authors describe an increased invasive capacity upon COX-2 KO, thus I suggest to investigate cellular invasiveness in PDX models by immunohistochemistry. In this respect, CD44 was already discussed as marker in mesenchymal glioma subpopulations. Staining of CD44 in orthotopic PDX models and spheroids would support this data.
Response 2: We created 3D images of the cells (attached figue). It can be seen that the COX-2 knockout cells appear to be larger and more amoeboid, with a flattened and enlarged nucleus compared to the U87 wild-type cells. Based on their structure in the monolayer studies, we assume that they have an advantage in migration and invasion. Furthermore, in this experimental approach, they are adequately supplied with nutrients and oxygen, and the inhibitory effects that we later observe in spheroid and tumor growth are masked. A more specific differentiation of the cell models studied here, including distinguishing between cells with a more or less mesenchymal phenotype, was beyond the scope of our manuscript. Referring to the reviewer's comment on the spheroid and animal experimental approaches, we would like to point out that all studies were conducted on the U87 model (wild-type versus COX-2 knockout). The possibility of investigating PDX models is beyond our capabilities (also in terms of animal testing approval) and is beyond the scope of this manuscript.
Comments 3: As the submitted study is solely based on one cell line, I suggest to demonstrate the effects of COX-2 on glioblastoma aggressiveness (hypoxia, proliferation, ROS induction etc.) by including more cell models. In case the establishment of additional CRISPR-KO-models is not feasible, this work would also benefit from treatment experiments with COX-2 inhibitors e.g Celecoxib in additional glioblastoma cells (spheroid assays). This data would further strengthen a clinical translation of COX-2 inhibition in glioblastoma.
Response 3: We agree with the reviewer's objection that, given the heterogeneity of various tumor entities, and this also applies to glioblastoma, it is desirable to investigate multiple cell lines. We deliberately chose the U87 model for our approach because it is very widespread and sufficiently characterized, making it easy to compare our results with data from other groups. Of course, no generalized conclusions for glioblastoma can be drawn from this. We have discussed this limitation with the following statement in the manuscript.
In the conclusion section, line 589-591, red marked
“The studies on the U87 cell model do not fully reflect the heterogeneity of the tumor entity. Accordingly, our observation represents a basic finding of a commonly used specific glioblastoma cell line rather than a generalizable finding with immediate specific clinical relevance to glioblastoma”.
We tested two selective COX-2 inhibitors (celecoxib and rofecoxib) on both cell models in a concentration- and time-dependent viability assay (attached figure). No differences were observed between cells with COX-2 and without COX-2 expression. Only very high concentrations of celecoxib and rofecoxib affected the cell viability, which, however, did not appear to be due to targeted COX-2-dependent effects. In summary, chemosensitivity in U87 glioblastoma cells is not affected and appears to be independent of COX-2 expression. Celecoxib and rofecoxib did not act via COX-2-dependent effects, and COX-2-independent signaling pathways have to be considered. The so called off-target-effects of celecoxib have been previously described by others. Here, binding affinity to proteins other than the COX-2 enzyme with relation to tumor metastasis and angiogenesis is highlighted. In our studies, we aimed to demonstrate that permanent inhibition of COX-2, i.e., a CRISPR/Cas9 induced knockout of COX-2, has metabolic and, in this sense, therapeutic effect.
Temporary inhibition, e.g. with siRNA or COX-2 specific inhibitors, exhibits only transient effects, despite all the side effects.

Reviewer 2 Report
Comments and Suggestions for Authors
In the manuscript by Jens Pietzsch et al. entitled “Exploring the role of peripheral macrophages in glioma pro-2 gression: Metabolic significance of Cyclooxygenase-2 (COX-2)” it is demonstrated that knocking out the COX-2 gene in glioblastoma cells significantly inhibits tumor growth and increased cell death.
The scientific team developing these studies and publication already has a good experience in the CRISPR/Cas9 technique and analysis of the impact of COX-2 in the context of another cancer - melanoma (published in 2022). Therefore, I treat this publication as a continuation of their scientific development.
The manuscript is well written, all sections are relevant, clear and make a significant contribution to the publication value.
Some edytorial/technical issues or concerns should be addressed:
1) the order of cited publications; line 58 has 9,10,11, can be changed to 9-11; then on line 60 there are references to works 11-13 and on line 64 already to works 17,18; I would like to ask you to check what about references to works 14,15,16 - I have the feeling that they are not cited anywhere. I kindly ask the authors to verify the numbering of citations.
2) in the caption of figure 1 there are different font formats; please unify.
3) sometimes there are blank lines between paragraphs, sometimes there aren't any; please standardize
4) there are many paragraphs with headings in the publication; using such frequent headings and such short text from my perspective makes it difficult to read the publication smoothly. I suggest to verify and remove some paragraph headings to make the text more fluid.
5) the position of the letters indicating the panels on the figures needs to be changed; the current format does not meet the standards, sometimes it is unclear which letter corresponds to which panel; the letter designation should be in the upper left corner of the figure, in a larger font, clearly visible
6) Figure 3- both tables must be removed from the figure; cited in the text as tables and placed as tables in the text; currently the lower-case letter reference (a) to both tables is inconsistent with the standards, and illegible.
7) Line 600, I could not open the supplementary file from the link; I verified its content based on the attached document from the MDPI system, but the link itself did not contain any materials - to be verified
Author Response
Comments 1: the order of cited publications; line 58 has 9,10,11, can be changed to 9-11; then on line 60 there are references to works 11-13 and on line 64 already to works 17,18; I would like to ask you to check what about references to works 14,15,16 - I have the feeling that they are not cited anywhere. I kindly ask the authors to verify the numbering of citations.
Response 1: Many thanks for the important information. A mistake has crept in here when citing. We have corrected the citation numbers in the introduction. We have also summarized the citations with more than 3 authors. We marked everything in red (line 56, 58, 64, 67, 327).
Comments 2: in the caption of figure 1 there are different font formats; please unify.
Response 2: Thank you for the comment, we have adjusted the font sizes.
Comments 3: sometimes there are blank lines between paragraphs, sometimes there aren't any; please standardize
Response 3: We have taken out the blank lines.
Comments 4: there are many paragraphs with headings in the publication; using such frequent headings and such short text from my perspective makes it difficult to read the publication smoothly. I suggest to verify and remove some paragraph headings to make the text more fluid.
Response 4: Thank you for this helpful comment. We have removed all the subheadings in the results section. We hope this improves readability.
We have adapted the headlines for the result section part 2.3. (line 179) and 2.4. (line 224) and marked this in red.
Comments 5: the position of the letters indicating the panels on the figures needs to be changed; the current format does not meet the standards, sometimes it is unclear which letter corresponds to which panel; the letter designation should be in the upper left corner of the figure, in a larger font, clearly visible
Response 5: Here we followed the rules of the input mask. However, we have made the panel letter designation larger and highlighted them in bold.
Comments 6: Figure 3- both tables must be removed from the figure; cited in the text as tables and placed as tables in the text; currently the lower-case letter reference (a) to both tables is inconsistent with the standards, and illegible.
Response 6: Many thanks for this comment. We have now removed the 2 tables from Figure 3 and placed them in the appropriate place (marked in red).
Table 1 line 185
Table 2 line 189
Comments 7: Line 600, I could not open the supplementary file from the link; I verified its content based on the attached document from the MDPI system, but the link itself did not contain any materials - to be verified
Response 7: I have uploaded the additional data. It may be that the journal itself creates the link.
